# Trends in Work Conditions and Associations with Workers’ Health in Recent 15 Years: The Role of Job Automation Probability

**DOI:** 10.3390/ijerph17155499

**Published:** 2020-07-30

**Authors:** Wan-Ju Cheng, Li-Chung Pien, Tomohide Kubo, Yawen Cheng

**Affiliations:** 1Department of Psychiatry, China Medical University Hospital, Taichung 40447, Taiwan; 2Department of Public Health, China Medical University, Taichung 40402, Taiwan; 3Post-Baccalaureate Program in Nursing, College of Nursing, Taipei Medical University, Taipei 11031, Taiwan; starsky5202001@gmail.com; 4National Institute of Occupational Safety and Health, Kawasaki 214-8585, Japan; tomohide.0911@jniosh.com; 5Institute of Health Policy and Management, Department of Public Health, National Taiwan University, Taipei 100, Taiwan; ycheng@ntu.edu.tw

**Keywords:** job automation, psychosocial work conditions, burnout, self-rated health, trend analysis

## Abstract

Job automation and associated psychosocial hazards are emerging workplace challenges. This study examined the trends in work conditions and associations with workers’ health over time in jobs with different automation probabilities. We utilized data from six waves of national questionnaire surveys of randomly selected 95,762 employees between 2001 and 2016. The Job Content Questionnaire, the Copenhagen Burnout Inventory, and the Self-Rated Health Scale were applied, and working time was self-reported. Automation probability was derived for 38 occupations and then categorized into three groups. Trends in work conditions and the associations between automation probability, work conditions and health were examined. We observed a 7% decrease in high automation probability jobs, an overall increase in job demands for and prevalence of shift work, and a decrease in job control. Workers with high automation probability jobs had low job demands, low job control and high job insecurity. Low automation probability was associated with burnout in logistic regression models. The odds ratio of job insecurity, long working hours, and shift work relating to health was higher in the later years of the surveys. In conclusion, there has been a decrease in high automation probability jobs. Workers employed in jobs with different levels of automation probability encountered different work condition challenges.

## 1. Introduction

Large scale epidemiological studies have tested models and established associations between adverse psychosocial work conditions and stress-related health [1]. Numerous studies have also examined changes in psychosocial work conditions, and some of them have indicated deteriorating trends in work quality over time. For instance, studies from western societies have reported rising precarious employment [2,3], increasing work demands, decreasing job control [4,5,6,7] and increasing prevalence of long working hours and irregular work shift [8]. Cheng and her colleagues have documented deteriorating trends in psychosocial work conditions among general workers in Taiwan during the period from 2001 to 2010, including a rising prevalence of long working hours and shift work and increasing trends in reduced job control levels [9]. On the contrary, some studies have indicated improvement in psychosocial work conditions. For example, a study from Denmark showed improvement in job security, job control and work shift arrangements and decline in low-skilled jobs between 1990 and 2000 [10], and a study from Canada showed improvement in workplace support, job security, regular work shifts, working hours and work demands during the period from 2002 to 2012 [11]. However, while the rise of automation in the workplace has been a worldwide phenomenon, few studies have been carried out to examine the trend in job automation probability and how such a trend might affect employment and workers’ psychosocial health risks.

Automation is eminent in Taiwan as well as in many other East Asian countries where labor-intensive manufacturing industries conglomerate. According to a report by the International Federation of Robotics, Taiwan ranks as the 10th most automated country in the world, with a robot installation rate increasing by 26% per year between 2012 and 2017 [12]. There are pros and cons of job automation. On the one hand, automation could help improve work efficiency and reduce workers’ exposure to unsafe or unpleasant work conditions. On the other hand, automation has the capability to decimate low-skilled jobs by replacing humans with robots, especially in regions with rising labor costs [13].

With regard to the assessment of automation probability, the measure developed by Frey and Osborne has been widely cited [14]. The scale was constructed to measure automation probability for each occupation based on three dimensions, including the degree of utilizing perceptive and manipulative abilities, the degree of creativity, and the level of social intelligence. These three dimensions were assessed by nine variables, which are thought to be the bottlenecks to automation, namely: finger dexterity, manual dexterity, cramped work space and awkward positions, originality, fine arts, social perceptiveness, negotiation, persuasion, and assisting and caring for others. Using this measure, Patel et al. found that USA workers in jobs with higher automation probability had greater job insecurity, which in turn was associated with poorer health [15]. This study supported the hypothesis that expectations of unemployment and reduced wages brought about by work automation increase workers’ perception of job insecurity. Reciprocally, a Norwegian study observed that employees with poor health were more likely to lose jobs due to job automation [16]. However, in these studies psychosocial conditions at work were not investigated, and their relationship with job automation has not been not examined.

Job automation is expected to replace human labor in routine tasks and increase job insecurity and workers’ psychological health risks. Yet, to our knowledge, the impacts of automation on employment and workers’ health risks have rarely been studied in East Asian working populations, and psychosocial work conditions have not been considered in this relationship. In this study, we utilized data from six waves of national surveys over 15 years and derived automation probability for each occupation based on the methods developed by Frey and Osborne. The aims of this study were to examine: (1) changes in work and health conditions by the level of automation probability over time, and (2) associations between automation probability and workers’ health status by survey year. As the negative impacts of automation on workers’ health have been increasingly noticed, we hypothesized that work and health conditions in jobs with high automation probability might deteriorate and the associations of job automation probability with workers’ health might be stronger in more recent years.

## 2. Methods

### 2.1. Study Populations

The Ministry of Labor of Taiwan has conducted nationwide surveys of the working population every 3–5 years since 1988 to understand safety and health conditions in the workplace. For each survey, participants were selected through a two-stage random sampling process. In the first stage, all districts and villages throughout Taiwan were grouped into strata according to their levels of urbanization. A random sample of districts and villages was chosen from each stratum. In the second stage, a random sample of households was selected within each district or village, and residents of the sampled households who were employed at the time of the survey were identified and invited to participate. Subjects who were not economically active were not eligible. Self-administered questionnaires were delivered to the selected households by trained interviewers. After one week, completed questionnaires were collected and on-site inspection was performed by the same interviewer. The response rates for the six waves of survey were 82%, 81%, 86%, 87%, 89%, and 78%, respectively. In this study, we utilized survey data collected in 2001, 2004, 2007, 2010, 2013, and 2016. We excluded employers, the self-employed, and people aged <25 or >65 years. A total of 95,762 participants were included in the study and it was carried out following the rules of the Declaration of Helsinki.

### 2.2. Measurements

Frey and Osborne derived automation probabilities for 702 occupations classified by the Standard Occupational Classification (SOC) of the United States Department of Labor [14]. In our study, participants’ occupations were coded and classified into 38 occupational groups according to the sixth edition of the Standard Occupational Classification by the Taiwan Ministry of Labor. This classification corresponds to the International Standard Classification of Occupations (ISCO-08) two-digit major and sub-major groups. For each of these occupational groups, an average automation probability was derived by matching the ICSO-08 occupational classification with the SOC. Occupations were then ranked and categorized into terciles of high, median and low automation probability groups.

Study participants were asked to provide information regarding their total working hours one week prior to the survey. These working hours were categorized into three groups: ≤40, 41–48, and >48 h per week. We defined long weekly working hours as >48 h per week. Participants self-reported their work time arrangement, with a response chosen from fixed work shift, rotating work shift and irregular work shift. Those who chose fixed shift were further asked if they worked in the late evening or nighttime. Shift work was defined as having a work shift other than fixed daytime work.

Job control and psychosocial job demands were assessed using the validated Chinese version of the Job Content Questionnaire (JCQ) based on the job strain model by Karasek and Theorell [17,18]. Due to the space limitation of the questionnaire, items were selected from the JCQ, and this measurement of job control and job demands with partial items has been proven to be valid [19]. Four items (work is fast, work is hectic, work is hard, must concentrate on the job for a long time) from the original five-item questionnaire for the demands scale and a seven-item subscale (learning new things, non-repetitive work, creative work, various tasks, can develop one’s abilities, freedom to make decisions, opinion is influential) from the original nine-item questionnaire for the control scale were included in the questionnaires in year 2004, 2007, 2013, and 2016. The items were listed as a statement with the response recorded on a four-point Likert scale ranging from 1 (strongly disagree) to 4 (strongly agree). The internal consistency for the seven-item job control scale and four-item demands scale were acceptable for 2004 (standardized Cronbach’s α = 0.75 and 0.74, respectively), 2007 (standardized Cronbach’s α = 0.72 and 0.73, respectively), 2013 (standardized Cronbach’s α = 0.70 and 0.79, respectively), and 2016 (standardized Cronbach’s α = 0.73 and 0.83, respectively) surveys. The mean scores for job control and demands were then ranked and divided into terciles (low, medium and high). One item for job insecurity (“my job is secure”) was assessed using a four-point Likert scale and its response was coded dichotomously (agree/disagree). Age, gender, and educational level were also included in the self-reported questionnaire.

Two health outcomes were assessed by the questionnaire. Self-rated health (SRH) is a composite indicator for universal dimensions of health and has been found to predict mortality [20]. SRH was assessed by a single-item question, “In general, how is your health?”, which had five possible answers: “very good”, “good”, “moderate”, “poor” and “very poor”. In this study, the responses were dichotomized into poor SRH (poor or very poor) or good SRH (very good, good, or moderate). Burnout is conceptualized as an affective reaction to prolonged work stress [21] and has been found to be associated with sickness absence, physical diseases and mental illnesses [22,23,24]. Burnout status was assessed by the five-item scale for personal burnout from the Copenhagen Burnout Inventory [25], which has been validated and used worldwide [26,27,28,29] to evaluate employees’ health status related to long-term involvement in emotionally demanding work [30]. The responses for five items were recorded on a five-point scale: “always” (score 100), “often” (75), “at times” (50), “not often” (25), and “never” (0), and a mean score >50 was classified as having burnout. However, the burnout scale was not included in the questionnaires used for the surveys of 2001 and 2010.

### 2.3. Statistical Analysis

Age, gender, educational level, work conditions and health indicators were examined for each of the job automation probability group by survey years. A linear-by-linear association test and general linear models were used to examine the trend of categorical variables and continuous variables of work conditions, respectively, over the studied period. Trend analysis was performed with IBM SPSS Statistics for Windows, Version 24.0 (Armonk, NY, USA). Logistic regression analysis was used to examine the associations between job automation probability and health, adjusting for work conditions for 2004, 2007, 2013, and 2016 separately. SAS 9.4 (SAS Institute, Cary, NC, USA) was used for the analyses. The significance level was set at *p* < 0.001.

## 3. Results

Table 1 shows the demographic characteristics of the three automation probability groups. Common job types of the high automation probability group were machine operators, assemblers, salespersons and metal operators; common job types of the medium automation probability group were business and administrative associates, construction workers and vehicle operators; and common job types of the low automation probability group were professionals and higher education teachers. Women and less educated workers were more prevalent in the high automation probability group as compared to those in the medium and low automation probability groups.

As shown in Figure 1, the percentage of workers with high automation probability jobs decreased by 7.2% (from 45.64% to 38.48%), while those with medium automation probability jobs increased by 7.4% (from 36.26% to 43.67%) over the studied period (details shown in the Appendix A). Overall, the average age of workers over the studied period increased by 3.3 years and the percentage of female workers increased by 2.7%. In trend analysis, we found significantly decreased total working hours and percentage of job insecurity, poor SRH and burnout, but increased percentage of long working hours (>48 per week) and shift work, and decreased job control scores (please refer to data shown in the Appendix A).

Figure 2 shows that working hours decreased slightly in all of the three groups, but the decrease was most prominent in workers with high automation probability jobs. However, trend analysis showed the number of workers with long working hours significantly increased in high and low probability groups. Overall, the percentage of shift workers increased by 6.8%, and the increase was most prominent in workers with low probability jobs, in which the percentage doubled from 10.33% in 2000 to 20.95% in 2016. Between 2004 and 2016, the average job demand scores increased significantly in low automation probability jobs, but no significant trend was observed in workers with median and high automation probability jobs. Job control scores decreased steadily and significantly in all the three groups. A significant trend of decreasing job insecurity was observed in the high and median probability groups but not in the low automation probability group.

As to health outcome, the percentage of workers who reported poor SRH significantly decreased in workers with low and high automation probability jobs, and the percentage of workers who reported burnout significantly decreased in all groups but remained higher in the low probability group than in the other two groups.

Multiple logistic regression models showed that, as compared with workers with low automation probability jobs, those with high automation probability jobs had higher risks of poor SRH in 2016 (Table 2) and had lower risks of burnout in all the survey years (Table 3). High job demands and job insecurity were associated with poor SRH and burnout in all the survey years. The odds ratio of long working hours for burnout increased over the survey years (from 1.09 to 1.76). Job insecurity was associated with poor SRH and the odds ratio increased from 2004 to 2016 (1.76 and 2.25, respectively).

## 4. Discussion

This is the first study to examine the trend in psychosocial work conditions and worker’s health over the past 15 years according to automation probability. We observed a significant decrease in high automation probability jobs. Workers in low automation probability jobs reported doubled shift work prevalence, decreased job control, and increased job demands. High automation probability jobs were associated with poor self-rated health while low probability jobs were associated with burnout. The odds ratio of job insecurity for poor health, and long working hours for burnout, increased over the study period.

Studies from both the European Union and the United States indicated that there had been a reallocation of middling manufacturing and routine office workers into either low-paid service occupations or high-paid professionals and managers in recent years, a process known as job polarization [31,32]. Yet our findings indicated that in Taiwan employment in high automation probability jobs had shrunk. However, while the growth of high-skilled jobs was observed in other studies, we did not observe an increase in low automation probability occupations. Such an inconsistency might be explained by the existence of a skill barrier or educational gap, barring low-skilled workers from moving upward to high-skilled jobs. This could also be explained by a mismatch between skills demanded in the labor market and skills acquired through higher education in Taiwan, as the rapid expansion of higher education has not contributed to industrial upgrading and greater demands for a high-skilled workforce, but has instead led to greater uncertainty in wage prospects [33]. The stagnation of growth in the high-skilled workforce could also be a result of labor market globalization, as observed in Latin American countries [34]. In the context of job automation, labor market globalization, job content changes, continuous on-site training of employees, and education renovation according to the demands of the workforce market are needed.

In addition, this study showed that psychosocial work conditions had deteriorated over time in all of the three automation probability groups, as working hours had decreased only minimally, but the prevalence of workers with prolonged working hours and non-standard work shifts had increased and the level of psychosocial job demands had increased substantially. Furthermore, deteriorating trends appeared to be more apparent in the low automation probability group, with the prevalence of shift work increased by two-fold. These findings suggested that while high automation probability jobs were likely to be decimated and replaced by robots, those with low automation probability jobs could also be affected in the process of technological innovation. One of the possibilities of such trends is the emergence of emotional aspects of psychological job demands, which are more prevalent in jobs involving more human interactions. These psychosocial stressors, including psychological and verbal violence [35], emotional demands and role conflicts originating from the interactions with clients [36,37], correspond closely to one of the three constructs of automation probability, i.e., the use of social intelligence. The development of an information-based economy may also contribute to the sharp increase in short-duration shift work and round-the-clock service, especially in human service sectors [38].

Another novel finding of the present study was that, along with a slight decrease in average working hours, the prevalence of workers with prolonged working hours had increased and the associations of prolonged working hours with poor health and burnout had become greater over time. These observations suggested a polarizing distribution of working hours, which was also observed in Taiwan [9] and in the United States [39]. Long working hours are known to contribute to work stress and stress-related health risks. Nevertheless, it is worth noticing that long working hours per se may not necessarily be associated with adverse health risks, because workers’ motives and self-control in working hours arrangement may differ greatly by social context [40,41]. The increasing association between long working hours and burnout over time may also be explained by these changes in social context including perceived economic uncertainty and social norms of ideal worker type or ideal working hours.

However, high automation probability jobs were associated with poor SRH only in 2016 but not in earlier surveys. We also observed an increase in the prevalence of poor SRH after 2010 among employees with high and median automation probability jobs. These findings were probably due to a selection effect as workers with ill health were more likely to have drifted to insecure and low-skilled jobs in more recent years. Poor health was found to be a predicting factor for working in high automation probability jobs in a Norwegian study [16]. It can also be anticipated that along with the trends in job automation, workers with poor health are exposed to an additional risk of job insecurity and unemployment. It is worth noticing that in our study job insecurity was found to have an increasing odds ratio for poor SRH over time, while workers in high automation probability jobs reported the highest percentage of job insecurity. The overall improvement of SRH and burnout scores in our study may also be a result of healthy worker selection on a larger scale, while workers with existing health problems had difficulties in staying active in the labor market.

This study has several limitations. First, although all the six waves of surveys consisted of a representative sample of employees, these participants were independently recruited for each survey and were not followed in succeeding surveys. Therefore, participants who had left work or the labor market due to health problems were not considered. This may contribute to a healthy worker effect leading to a healthier working population and attenuated associations between adverse work conditions and poor health in later years. Other unobserved changes in characteristics of workers in each occupational group may also lead to selection bias. Secondly, the automation probability of jobs may not be the same in the United States as in Taiwan. The skills needed in specific jobs and the level of automation in specific industries differ worldwide. Furthermore, we had access only to the first two digits of occupational code of the classification system for each participant; therefore, the averaged automation probability for two-digit occupational groups from the probability estimated for six-digit occupations by Frey ad Osborne [14] may not be representative for the identified 38 occupational groups. Furthermore, heterogeneity in each of the 38 occupational groups has been neglected. For example, salespersons were classified as a high-automation probability job, but the requirement for social perceptiveness, negotiation, and persuasion differs between different types of salesperson. The survey participants did not include foreign workers, and studies concerning work conditions and health and safety of foreign workers were limited to occupational injuries [42]. Nevertheless, a substantial proportion of foreign workers worked in the service sector, e.g., as caregivers [43]. Future studies should include the growing foreign worker population. Thirdly, health conditions and work characteristics measurements were based on self-report and were subject to recall bias or social desirability bias.

## 5. Conclusions

The findings of this study showed that jobs with high automation probability had decreased over the studied period, and workers employed in jobs with different levels of automation probability encountered different types of psychosocial work hazards and health risk. For workers whose employment is vulnerable to automation, employment policies should be improved to ensure continuous on-job training and skill development according to the demands of the workforce market. Along with the trend towards automation, social policies should also be improved to ensure more equal distribution of economic gains, in order to protect workers whose health status makes them less competitive in the current labor market.

Furthermore, the fact that all workers are likely to be affected by the trend towards job automation deserves further investigation. Especially for workers who have to engage in intense social interactions with clients, workplace policies should be developed to reduce the impacts of emotional demands and human-machine interactions on workers’ burnout. For instance, abundant studies have found that an increase in job resources and work engagement help attenuate burnout [44,45,46,47], and an improvement in organizational psychosocial climate has been found to help decrease modern psychosocial stressors and negative health consequences [48,49,50]. The increasing demand for shift workers should also be reexamined, and extremely irregular or long/short shift work contracts should be regulated by the government. Economic studies are needed to seek a balance between 24-h service utility and the cost to shift workers’ health. The increasing association between long working hours and burnout over time may have reflected a changing social norm of work-life balance. With higher working hours compared to other industrialized countries, the organizational culture of overtime work should be challenged in Taiwan.

## Figures and Tables

**Figure 1 ijerph-17-05499-f001:**
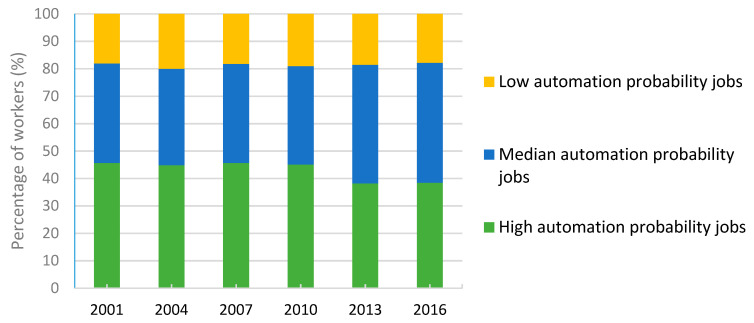
Temporal trends in number of workers in three levels of automation probability jobs. Number of participants in the six waves of survey were 14,691, 15,288, 17,042, 17,263, 16,530, and 14,948, respectively.

**Figure 2 ijerph-17-05499-f002:**
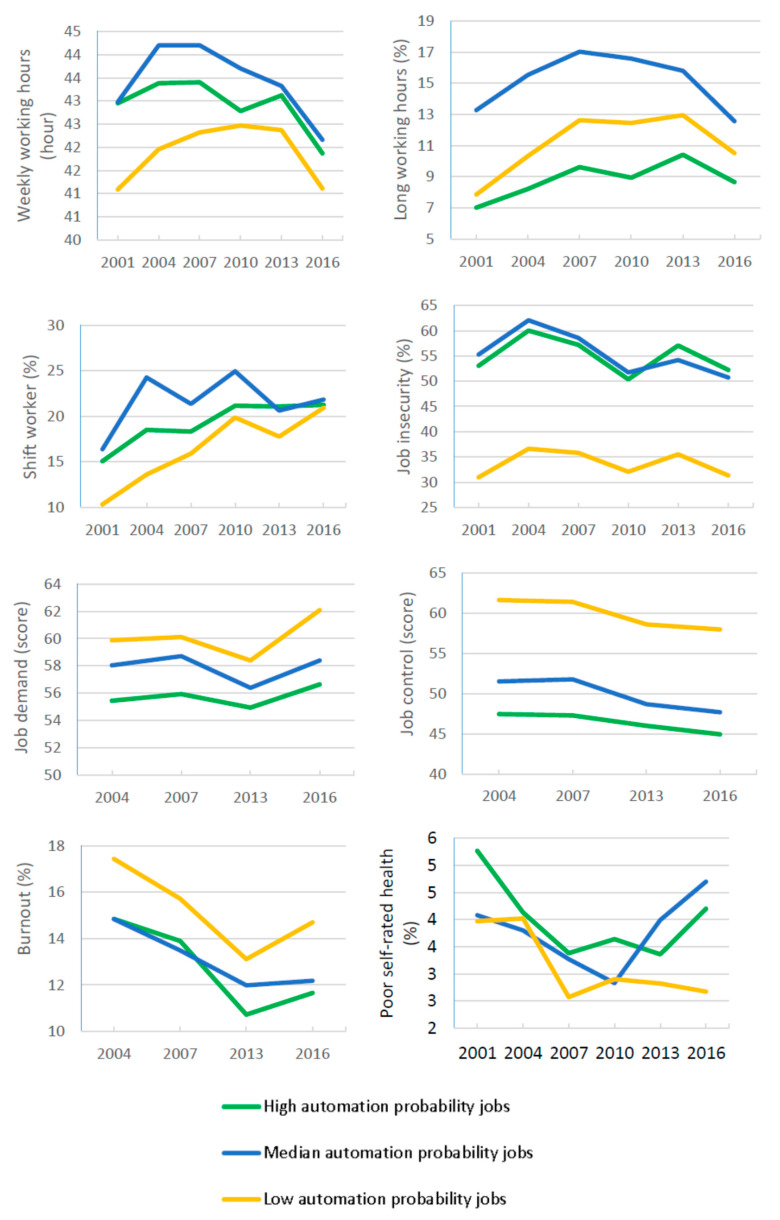
Temporal trends in work conditions, poor self-rated health, and burnout stratified by three levels of job automation probability.

**Table 1 ijerph-17-05499-t001:** Demographic characteristics of employees of high, median, and low automation probability jobs in all six waves of survey (*n* = 95,762).

	High Automation Probability Jobs (*n* of Jobs = 13, *n* of Workers = 41,199)	Median Automation Probability Jobs (*n* of Jobs = 13, *n* of Workers = 36,705)	Low Automation Probability Jobs (*n* of Jobs = 12, *n* of Workers = 17,858)
Examples of Jobs	Machine operator and assemblers (23%) Salespersons (14%) Metal operators and preparers (13%)	Business and administrative associates (29%) Construction workers (12%) Vehicle operators (12%)	Science and engineer professionals (25%) Higher education teachers (18%) Medical service professionals (14%)
Age (years)	39.74 ± 9.83	40.51 ± 9.86	39.66 ± 9 ± 46
Gender (Female)	57.62%	27.53%	42.03%
Education			
Primary Education	10.96%	8.42%	0.55%
Secondary Education	72.33%	72.79%	31.30%
University and Above	16.71%	18.80%	68.14%

**Table 2 ijerph-17-05499-t002:** Odds ratio and 95% confidence interval (95% CI) for poor self-rated health in the survey years, *n* = 15,288, 17,042, 16,530, and 14,948, respectively.

	2004 (Case = 608)	2007 (Case = 544)	2013 (Case = 576)	2016 (Case = 618)
	OR (95% CI)	OR (95% CI)	OR (95% CI)	OR (95% CI)
Age >55	1.26 (0.87, 1.83)	1.91 (1.43, 2.54) *	1.91 (1.51, 2.41) *	1.80 (1.45, 2.24) *
Gender (reference: female)	1.11 (0.93, 1.32)	1.07 (0.89, 1.29)	1.05 (0.89, 1.25)	1.09 (0.92, 1.28)
Automation probability				
Low	1	1	1	1
Median	0.84 (0.66, 1.07)	1.06 (0.81, 1.40)	1.09 (0.84, 1.42)	1.45 (1.10, 1.91) *
High	0.91 (0.72, 1.16)	1.11 (0.85, 1.46)	0.92 (0.70, 1.21)	1.34 (1.01, 1.77) *
Working hours				
≤40	1	1	1	1
40 < hours ≤ 48	0.79 (0.66, 0.95) *	0.69 (0.52, 0.92) *	0.86 (0.70, 1.05)	0.92 (0.75, 1.12)
>48	1.03 (0.79, 1.33)	0.90 (0.63, 1.25)	1.20 (0.95, 1.52)	1.38 (1.09, 1.76) *
Shift work	1.16 (0.95, 1.41)	1.22 (0.99, 1.50)	1.42 (1.17, 1.73) *	1.41 (1.17, 1.69) *
Job demand				
Low	1	1	1	1
Median	1.22 (0.95, 1.57)	0.99 (0.77, 1.29)	1.18 (0.90, 1.55)	0.98 (0.73, 1.31)
High	1.94 (1.60, 2.35) *	1.48 (1.22, 1.79) *	2.01 (1.67, 2.43) *	1.98 (1.64, 2.39) *
Job Control				
High	1	1	1	1
Median	1.25 (1.00, 1.56) *	1.18 (0.93, 1.49)	1.55 (1.22, 1.97) *	1.33 (1.05, 1.69) *
Low	1.05 (0.86, 1.29)	1.06 (0.85, 1.32)	1.03 (0.82, 1.31)	1.18 (0.93, 1.48)
Job insecurity	1.76 (1.46, 2.11) *	1.89 (1.56, 2.30) *	1.92 (1.59, 2.33) *	2.25 (1.88, 2.70) *

* *p* < 0.001.

**Table 3 ijerph-17-05499-t003:** Odds ratio and 95% confidence interval (95% CI) for burnout in the survey years, *n* = 15,288, 17,042, 16,530, and 14,948, respectively.

	2004 (Case = 2341)	2007 (Case = 2399)	2013 (Case = 1910)	2016 (Case = 1854)
	OR (95% CI)	OR (95% CI)	OR (95% CI)	OR (95% CI)
Age >55	0.70 (0.54, 0.92) *	0.77 (0.62, 0.96) *	0.92 (0.77, 1.11)	1.00 (0.84, 1.18)
Gender (reference: female)	1.56 (1.42, 1.72) *	1.49 (1.36, 1.64) *	1.45 (1.31, 1.60) *	1.43 (1.29, 1.59) *
Automation probability				
Low	1	1	1	1
Median	0.85 (0.75, 0.97) *	0.82 (0.72, 0.93) *	0.91 (0.79, 1.04)	0.83 (0.72, 0.96) *
High	0.83 (0.73, 0.95) *	0.82 (0.72, 0.93) *	0.77 (0.66, 0.89) *	0.7 (0.68, 0.92) *
Working hours				
≤40	1	1	1	1
40 < hours ≤ 48	0.93 (0.84, 1.02)	1.02 (0.85, 1.22)	1.09 (0.98, 1.23)	1.03 (0.91, 1.16)
>48	1.09 (0.95, 1.27)	1.22 (1.00, 1.50) *	1.31 (1.14, 1.51) *	1.76 (1.52, 2.03) *
Shift work	1.19 (1.06, 1.33) *	1.33 (1.20, 1.48) *	1.24 (1.10, 1.40) *	1.29 (1.15, 1.45) *
Job demand				
Low	1	1	1	1
Median	1.93 (1.66, 2.23) *	1.81 (1.57, 2.08) *	1.96 (1.66, 2.31) *	1.98 (1.65, 2.38) *
High	4.09 (3.65, 4.60) *	3.50 (3.13, 3.90) *	4.19 (3.71, 4.74) *	4.47 (3.90, 5.12) *
Job Control				
High	1	1	1	1
Median	0.79 (0.70, 0.90) *	0.99 (0.87, 1.12)	1.04 (0.91, 1.20)	0.89 (0.78, 1.03)
Low	0.87 (0.78, 0.97) *	1.04 (0.93, 1.16)	1.04 (0.91, 1.17)	0.96 (0.85, 1.09)
Job insecurity	1.89 (1.71, 2.09) *	1.63 (1.48, 1.80) *	1.54 (1.38, 1.71) *	1.80 (1.62, 2.00) *

* *p* < 0.001.

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
