# Peer review of "Trends in Work Conditions and Associations with Workers’ Health in Recent 15 Years: The Role of Job Automation Probability"

_ijerph, 2020, doi:10.3390/ijerph17155499_

Round 1

Reviewer 1 Report

     In this paper, trends of work conditions and the associations between automation probability and work conditions and health were examined. I think the following suggestions are meaningful for further improvement of the article:

  1. This article does not have a literature review section, nor does it have a general description of previous research in the introduction section. And there is no emphasis on the innovation of this study.
  2. Both the introduction and keyword sections of the article mention the research on workers' psychosocial health, but the study only uses job burnout as the basis for workers' psychosocial health, and the article does not indicate which existing studies can support this view.
  3. The typographical problems of Table 2 and Table 3 in the article are serious, and I cannot understand which numbers correspond to which variables
  4. The supplementary table mentioned in the article cannot be found.

Author Response

Reviewer 1

In this paper, trends of work conditions and the associations between automation probability and work conditions and health were examined. I think the following suggestions are meaningful for further improvement of the article:

Author response

We sincerely thank for your effort and time for reviewing our article. In the following section, we addressed your comments point-by-point.

  1. This article does not have a literature review section, nor does it have a general description of previous research in the introduction section. And there is no emphasis on the innovation of this study.

Author response:

There have been few previous researches of the study topic, i.e., the association between automation and psychosocial work conditions. Therefore, in the first paragraph of the Introduction section, we reviewed the changing trends of psychosocial work conditions in different countries. To our knowledge, there has been only two studies directly examined the association between automation probability and health. Therefore, as you suggested, we have summarized their findings in the 3rd paragraph of the Introduction section. We also highlighted the innovation of this study at the end of this paragraph and the 4th paragraph:

“Using the measure, Patel et al had found that US workers in jobs with higher automation probability had greater job insecurity, which in turn was associated with poorer health [15]. This study supported the hypothesis that expectations of unemployment and reduced wages brought about by work automation increase workers’ perception of job insecurity. Reciprocally, a Norwegian study observed that employees with poor health were more likely to lose jobs due to job automation [16]. However, in these studies work psychosocial conditions were not investigated, and the relationship with job automation have not been not examined.

Job automation is expected to replace human labor in routine tasks and increase job insecurity and workers’ psychological health risks. Yet to our knowledge, the impacts of automation on employment and workers’ health risks have rarely been studied in East Asian working populations, and psychosocial work conditions have not been considered in the relationship. In this study”…

  1. Both the introduction and keyword sections of the article mention the research on workers' psychosocial health, but the study only uses job burnout as the basis for workers' psychosocial health, and the article does not indicate which existing studies can support this view.

Author response:

In this study we used two health indicators, the SRH (self-rated health) and burnout scales. Psychosocial work conditions include in this study are long working hours, shift work, job control, job demands, and job insecurity. Personal burnout was evaluated by the validated Copenhagen Burnout Inventory, which has been widely used in studies concerning employee’s health in response to long-term involvement in emotionally demanding situations. We have revised the last paragraph of 2.2. Measurements:

“Burnout status was assessed by the five-item scale for personal burnout from the Copenhagen Burnout Inventory [25], which has been validated and used worldwide [26-29] to evaluate employees’ health status relate to long-term involvement in emotionally demanding work [30]”.

  1. The typographical problems of Table 2 and Table 3 in the article are serious, and I cannot understand which numbers correspond to which variables

Author response:

We are sorry about the typographical problems, but it was not the same as our original file. We will inform the editorial office about this problem.

  1. The supplementary table mentioned in the article cannot be found.

Author response:

The supplementary table has been uploaded as a separate file to the main text as the editorial office required. In the revised version, we put Table S1 at the end of the main file for your reference.

Reviewer 2 Report

This report studied how the degree of automation probability affects parameters such as overtime, workshift stress, employee's self-rated health conditions and job insecurity etc. over a period of more than 15 years in Taiwan. The results are illustrative, although not unexpected. I have the following comments:

  1. page 3, first line, should it be "older" than 65?
  2. I wonder if some types of salespersons should not be classified into the high automation prob. group, since quite an amount of interpersonal and communication skill as well as improvisations are needed.
  3. How to explain 2010 re-bounce in poor SRH?
  4. I am aware of the significant injection of foreign labor (eg. from Vietnam and Indonesia) into Taiwan industry in the past 2 decades. In the six waves of survey, are these foreign workers, presumably more in the high and median automation prob. groups, included in the interviews and data collections? They have different cultural background which may affect their perceptions of burnout and job insecurity etc etc..... These should be discussed                                                              

Author Response

This report studied how the degree of automation probability affects parameters such as overtime, workshift stress, employee's self-rated health conditions and job insecurity etc. over a period of more than 15 years in Taiwan. The results are illustrative, although not unexpected. I have the following comments

Author response

We sincerely thank for your effort and time for reviewing our article. In the following section, we addressed the comments point-by-point.

  1. page 3, first line, should it be "older" than 65?

Author response:

Thanks for your kind reminder. We have revised the sentence as:

“We excluded employers, the self-employed, and people aged < 25 or > 65 years”.

  1. I wonder if some types of salespersons should not be classified into the high automation prob. group, since quite an amount of interpersonal and communication skill as well as improvisations are needed.

Author response:

We agree with you that some types of salespersons job require high-level of interpersonal and communication skills and should be moved to median or low automation probability jobs. Unfortunately, in our data from the 6th edition of the Standard Occupational Classification by the Ministry of Labor of Taiwan, we were unable to differentiate different types of salespersons. This has been included as a study limitation in the Discussion section:

Furthermore, heterogeneity in each of the 38 occupational groups has been neglected. For example, salespersons were classified as a high-automation probability job, but the requirement for social perceptiveness, negotiation, and persuasion varies between different types of salespersons”.

  1. How to explain 2010 re-bounce in poor SRH?

Author response:

In figure 2, a rebounce in poor SRH was observed in high- and median-automation probability jobs, but not in low automation probability jobs. In the 5th paragraph of the Discussion section, we have revised the content and postulated our hypothesis that employees with poor health conditions may drift to high- and median-automation probability jobs in the context of job automation:

“However, high automation probability jobs were associated with poor SRH only in 2016 but not in earlier surveys. We also observed an increase in the prevalence of poor SRH after 2010 among employees with high and median automation probability jobs. These findings were probably due to a selection effect as workers with ill health were more likely to be drifted to insecure and low-skilled jobs in more recent years”…

  1. I am aware of the significant injection of foreign labor (eg. from Vietnam and Indonesia) into Taiwan industry in the past 2 decades. In the six waves of survey, are these foreign workers, presumably more in the high and median automation prob. groups, included in the interviews and data collections? They have different cultural background which may affect their perceptions of burnout and job insecurity etc etc..... These should be discussed

Author response:

We agree with you that injection of foreign labor is an important workplace issue concerning their health and work conditions not only in Taiwan but also many industrialized countries. According to Ministry of Labor statistics, foreign labor accounted for approximately 5.7% of the total working population in Taiwan. Nevertheless, the 6-waves of survey applied to only employees in the Taiwanese Household Registration System. The reciprocal relationship between job automation and injection of foreign labor, as well as foreign labors’ health and work conditions deserve thorough research. We have included the omission of the foreign labor population as a limitation of this study:

The survey participants did not include foreign workers, and studies concerning work conditions and health and safety of foreign workers were limited to occupational injuries[42]. Nevertheless, a substantial proportion of foreign workers worked in the service sector, e.g., as caregivers[43]. Future studies should include the growing foreign worker population”.

Reviewer 3 Report

Article is of interest to IJREPH readers and I would only recomend minor changes.

Abstract could include some of the instruments used, such as the Copenhagen Burnout Inventory, and it can be reduced to non essential things. 

The first paragraph of the Discussion is more appropiate for Conclussions, since it repeats results without comparing with previous studies.

The conclussions only come from their results of the study. There are comments that coud fit, in any case, in the Discussion section. 

Author Response

Article is of interest to IJREPH readers and I would only recomend minor changes.

Author response

We sincerely thank for your effort and time for reviewing our article. In the following section, we addressed the comments point-by-point.

  1. Abstract could include some of the instruments used, such as the Copenhagen Burnout Inventory, and it can be reduced to non essential things.

Author response:

We have revised the method section of the Abstract as your recommendation:

The Job Content Questionnaire, the Copenhagen Burnout Inventory, and Self-Rated Health Scale were applied, and working time was self-reported”.

  1. The first paragraph of the Discussion is more appropiate for Conclussions, since it repeats results without comparing with previous studies. The conclussions only come from their results of the study. There are comments that coud fit, in any case, in the Discussion section.

Author response:

Thanks for your recommendations and we have revised the first paragraph of the Discussion section to avoid repeating the results. The comparison of our findings with previous studies were presented in the next paragraphs of the Discussion section. We have also revised the 5. Conclusion section by summarizing our discussion.

The first paragraph of Discussion:

“This is the first study to examine the trend of psychosocial work conditions and worker’s health over the past 15 years according to automation probability. We observed a significant decrease in high automation probability jobs. Workers in low automation probability jobs reported doubled shift work prevalence, decreased job control, and increased job demands. High automation probability jobs were associated with poor self-rated health while low probability jobs were associated with burnout. The odds ratio of job insecurity for poor health, and long working hours for burnout increased over the study period”.

The Conclusion section:
“Findings of this study showed that jobs with high automation probability had decreased over the studied period, and workers employed in jobs with different levels of automation probability encountered different types of psychosocial work hazards and health risk. For workers whose employment is vulnerable to automation, employment policies should be improved to ensure continuous on-job training and skill development according to the demands of workforce market are needed. Along with the trend of automation, social policies should also be improved to ensure more equal distribution of economic gains, to protect workers whose health status makes them less competitive in current labor market.

Furthermore, the facts that all workers are likely to be affected by the trend of job automation deserve further investigation. Especially for workers who have to engage in intense social interactions with clients, workplace policies should be developed to reduce the impacts of emotional demands and human-machine interactions on workers’ burnout. For instance, abundant studies have found that increase in job resources and work engagement help attenuate burnout [44-47], and improvement in organizational psychosocial climate has been found to help decrease modern psychosocial stressors and negative health consequences [48-50]. The increasing demand for shift workers should also be reexamined, and extremely irregular or long/short shift work contracts should be regulated by the government. Economic studies are needed to seek a balance between 24-hour service utility and the cost of shift workers’ health. The increasing association between long working hours and burnout over time may have reflected a changing social norm of work-life balance. With a higher working hour compared to other industrialized countries, organizational cultures of overtime work should be challenged in Taiwan”.